# New Strategies for the Treatment of Neuropsychiatric Disorders Based on Reelin Dysfunction

**DOI:** 10.3390/ijms23031829

**Published:** 2022-02-06

**Authors:** Yumi Tsuneura, Tsuyoshi Nakai, Hiroyuki Mizoguchi, Kiyofumi Yamada

**Affiliations:** 1Department of Cellular Pathology, Institute for Developmental Research, Aichi Developmental Disability Center, Kasugai 480-0392, Japan; ytsuneura@inst-hsc.jp; 2Department of Neuropsychopharmacology and Hospital Pharmacy, Nagoya University Graduate School of Medicine, Nagoya 466-8560, Japan; t-nakai@med.nagoya-u.ac.jp (T.N.); hmizoguchi@med.nagoya-u.ac.jp (H.M.); 3Department of Advanced Medicine, Nagoya University Hospital, Nagoya 466-8560, Japan; 4Medical Interactive Research and Academia Industry Collaboration Center, Research Institute of Environmental Medicine, Nagoya University, Nagoya 464-8601, Japan

**Keywords:** reelin, neuropsychiatric disorders, development, ADAMTS-3

## Abstract

Reelin is an extracellular matrix protein that is mainly produced in Cajal-Retzius cells and controls neuronal migration, which is important for the proper formation of cortical layers in the developmental stage of the brain. In the adult brain, Reelin plays a crucial role in the regulation of *N*-methyl-D-aspartate receptor-dependent synaptic function, and its expression decreases postnatally. Clinical studies showed reductions in Reelin protein and mRNA expression levels in patients with psychiatric disorders; however, the causal relationship remains unclear. Reelin-deficient mice exhibit an abnormal neuronal morphology and behavior, while Reelin supplementation ameliorates learning deficits, synaptic dysfunctions, and spine loss in animal models with Reelin deficiency. These findings suggest that the neuronal deficits and brain dysfunctions associated with the down-regulated expression of Reelin are attenuated by enhancements in its expression and functions in the brain. In this review, we summarize findings on the role of Reelin in neuropsychiatric disorders and discuss potential therapeutic approaches for neuropsychiatric disorders associated with Reelin dysfunctions.

## 1. Introduction

Reelin is an extracellularly secreted glycoprotein that is necessary for brain development and neuronal function. In the developing brain, Reelin is produced by Cajal-Retzius cells, which are mainly present on the surface of the neocortex [1,2]. After birth, Cajal-Retzius cell numbers markedly decrease, and Reelin is mainly synthesized in γ-aminobutyric acid (GABA)-ergic neurons in the hippocampus and cortex [3,4]. Reelin consists of an N-terminal region containing a secretory signal, an eight Reelin repeats (Reelin repeats), and a C-terminal region rich in basic amino acids [5]. Secreted Reelin binds to apolipoprotein E receptor 2 (ApoER2) and very low-density lipoprotein receptors (VLDLR) expressed on neuronal membranes via the fifth and sixth Reelin repeats [6,7], stimulates Src family tyrosine kinases (SFKs), such as Fyn and Src, and promotes the tyrosine phosphorylation of intracellular Dab1 [8,9]. Phosphorylated Dab1 activates the downstream pathway and promotes neurite growth, dendritic spine growth, and neuronal migration [10,11,12]. The down-regulated expression of Reelin is clinically associated with neuropsychiatric disorders, such as schizophrenia, autism spectrum disorder (ASD), and Alzheimer’s disease (AD) [13]. In this review, we introduce the molecular functions of Reelin in neurons. We also summarize research on the involvement of Reelin in neuropsychiatric disorders and discuss potential therapeutic approaches for neuropsychiatric disorders associated with Reelin dysfunctions.

## 2. Roles of Reelin in Neural Functions

### 2.1. Neuronal Migration and Cortical Development

The cerebral cortex in the early developmental stage consists of a layer called the preplate and the ventricular zone at which new neurons are produced. Neurons generated in the ventricular zone enter the preplate, which separates into marginal zones and subplates, and then migrate radially throughout the subplate, but stop just before the marginal zone. In the cerebral cortex, early-born neurons are placed on the ventricular side and late-born neurons on the superficial side, resulting in a proper cortical layer [14]. Previous studies reported that Reelin signaling plays a role in the correct migration of neurons during the developmental formation of the cerebral cortex [5,15,16,17]. A decrease in the secretion of the Reelin protein from Cajal-Retzius cells causes a major reversal of the layered structure of the cerebral cortex [5]. Several proteins have been identified as key molecules for Reelin-dependent neuronal migration. The activation of integrin α5β1 through the intracellular Dab1-Crk/CrkL-C3G-Rap1 pathway after Reelin binds to its receptors is required for cell body translocation at the end of the migration of cortical neurons [18]. Cofilin, an actin-binding protein, and Reelin cooperate to regulate cytoskeletal dynamics during neuronal migration [19]. ApoER2 and VLDLR are well known to be major receptors involved in neuronal migration via Reelin signaling [20,21]. ApoER2, a Reelin-binding receptor, controls several processes in neuronal migration during cortical development, such as the early stage of radial migration and the termination of migration [20]. In neonatal ApoER2 knockout (KO) mice, cortical neurons overmigrate into the marginal zone [20]. A major role for VLDLR is its suppression of neuronal invasion within the marginal zone during neocortical development [21]. Therefore, Reelin and its downstream signals play important roles in cerebral cortex formation by regulating neuronal migration during the development stage.

### 2.2. Neurite Outgrowth

Previous studies reported that dendrite complexity was significantly reduced in *reeler* mice carrying a homozygous *Reln* gene deletion, as well as in heterozygous *reeler* mice [10,22]. The levels of the dendrite-specific microtubule-associated protein (MAP2) were significantly reduced in the hippocampus of homozygous *reeler* mice and, to a slightly lesser extent, in that of heterozygous *reeler* mice. A treatment with a CR50 antibody, which blocks the biological functions of Reelin, significantly reduced dendrite length and the complexity of cultures from heterozygous *reeler* mice [10]. Reelin may accelerate hippocampal dendrite development through the VLDLR/ApoER2-Dab1 pathway [10]. Kupferman et al. reported that Reelin exerted its functions through downstream intracellular Dab1 and Src family tyrosine kinase (SFK) signaling cascades and regulated dendritic outgrowth [23]. The phosphatidylinositol 3-kinase (PI3K)-Akt-mammalian target of rapamycin (mTor)-S6 kinase 1 pathway, which is downstream of Reelin, is associated with the regulation of dendritic growth and cortical development [24]. Reelin-Dab1 signaling and serine/threonine kinase 25 (STK25) competitively regulate Golgi morphology and neuronal polarity, which is important for dendrite formation [25]. Recent findings suggested that similar to STK25, mammalian sterile 20-like kinase-3 (MST3), a member of the germinal center kinase III (GCKIII) subfamily, regulates neuronal migration and polarization in a mutually compensatory manner [26]. Chondroitin sulfate proteoglycans inhibit axonal elongation; however, this is canceled by the activation of the Reelin signaling pathway [27]. Cytoplasmic linker-associated protein 2 (CLASP2) is a plus-end tracking protein that specifically accumulates at the growth cone and is a cytoskeletal effector of the Reelin signaling pathway [28]. A treatment with Reelin increased the axon length of primary cultured hippocampal neurons, whereas CLASP2 small hairpin RNA decreased axon length. The treatment with Reelin did not affect axon length in neurons with the knockdown of CLASP2. Furthermore, the phosphorylation of CLASP2 may be necessary for Dab1 interactions and neurite outgrowth [28]. These findings suggest that Reelin promotes neurite development and also that the disruption of Reelin signaling may result in an abnormal neurite morphology.

### 2.3. Spine Formation

A treatment with Reelin has been shown to significantly increase dendritic spine density in primary cultured hippocampal neurons [29]. In addition, Reelin may increase the puncta numbers of synaptophysin and postsynaptic density protein 95 (PSD95). Moreover, the Ca^2+^/calmodulin-dependent protein kinase II β subunit may be necessary for the effects of Reelin on dendritic spine density [29]. Spine density on layer II/III in the prelimbic area of the prefrontal cortex (PFC) was lower in juvenile heterozygous *reeler* mice than in their wild-type littermates, and this decrease was attributed to the selective loss of spines with a small head diameter [30]. A reduction in dendritic spine density was also observed in the hippocampal pyramidal neurons of heterozygous and homozygous *reeler* mice. In hippocampal slice cultures of homozygous *reeler* mice, reduced spine density was restored by a treatment with Reelin [11]. Reelin supplementation may increase the spine density of hippocampal CA1 pyramidal neurons in wild-type mice, but not in ApoER2 KO mice [31]. ApoER2 and VLDLR are required for Reelin-induced dendritic spine formation [10,32]. Moreover, Dab1 and SFK activities may be necessary for the development of a normal dendritic spine density in organotypic hippocampal cultures [11].

### 2.4. Synaptic Function

A stimulation with Reelin was found to activate ApoER2 and VLDLR at excitatory synapses, and this was followed by increases in Dab1 phosphorylation and the activation of Src. This process promoted the linking of Src to PSD95. As a consequence, the tyrosine phosphorylation of the *N*-methyl-D-aspartate receptor (NMDAR) subunit physically associated with PSD95 increased, thereby promoting the activation of NMDARs [13,33,34]. Reelin organizes the regulation of the subunit composition of synaptic NMDARs and controls the surface mobility of the NR2B subunits of NMDARs. A previous study demonstrated that blocking the function of Reelin prevented maturation-dependent reductions in NR1/NR2B-mediated synaptic currents [35]. In heterozygous *reeler* mice, the expression levels of PSD95, NR2A, and NR2B were reduced in a postsynaptic density fraction [36]. Reelin increased the tyrosine phosphorylation of the NR2B subunit and enhanced glutamate-stimulated Ca^2+^ influx through NMDARs, suggesting that it physiologically regulates NMDAR activity [22,37]. Furthermore, Reelin may enhance the activity of the α-amino-3-hydroxy-5-methyl-4-isoxazolepropionic acid receptor (AMPAR) by PI3K-dependent surface insertion [34]. A perfusion with Reelin was found to enhance long-term potentiation (LTP) in the hippocampus of wild-type mice, but not in ApoER2 KO mice or VLDAR KO mice, which indicates that ApoER2 and VLDLR are required to enhance synaptic transmission in the hippocampus [38].

## 3. Reelin and Neuropsychiatric Disorders

As mentioned above, Reelin has many effects on brain formation as well as on morphological changes in the neuronal network, and thus its dysfunction may cause various brain-related diseases. In this paragraph, we discuss reports of mutations in the *RELN* gene in humans with neuropsychiatric disorders.

### 3.1. Schizophrenia

Schizophrenia is a mental disorder that presents with various symptoms, such as hallucinations, delusions, abnormal behavior, decreased motivation, and cognitive deficits. The first study on Reelin abnormalities in schizophrenia revealed that *RELN* mRNA, which encodes Reelin, and Reelin protein expression levels were significantly lower in the postmortem brains of patients with schizophrenia than in non-psychiatric subjects [39]. Furthermore, the expression levels of *RELN* mRNA in the whole blood of untreated schizophrenia patients were significantly lower than those in healthy controls [40]. *RELN* mRNA expression levels were elevated in patients with schizophrenia by a 12-week treatment with olanzapine, an antipsychotic, suggesting that alterations in *RELN* mRNA expression levels are associated with the effects of antipsychotic treatment [40]. In recent years, genome-wide association studies and meta-analyses indicated that rs7341475 and rs262355 genetic polymorphisms in the *RELN* gene correlated with the onset of schizophrenia [41,42,43]. The missense variation c.9575 C > G (p.T3192S) in *RELN* was identified by whole-exome sequencing with samples from three affected individuals and one unaffected individual in a Chinese family with schizophrenia [44]. A *de novo* copy number variant (CNV) analysis of Japanese schizophrenia patients recently revealed a new pathogenic deletion (12.6 kb) in *RELN* (*RELN*-del) [45]. Taken together, these findings indicate that changes in Reelin expression and genetic variations are risk factors for schizophrenia. In exon-targeted resequencing using next-generation sequencing technology, rare variants of the *DAB1* gene (p.G382C and p.V129I) were detected in patients with schizophrenia. Furthermore, these mutants of the Dab1 protein were more unstable than the wild-type protein, which may diminish Reeln-Dab1 signaling and contribute to the pathology of schizophrenia [46].

### 3.2. ASD

ASD is a developmental disorder characterized by abnormalities in social interactions and communication, localized patterns of interest, and repetitive behavior. Some mutations in the *RELN* gene (p.R1742Q, p.R1742W, p.R2290C, p.R2290H, p.R2292C, p.R2639H, and p.R2833S) have been identified in patients with ASD [47,48,49,50]. In a whole-genome sequencing analysis, heterozygous variants of the *RELN* gene (p.S630R and p.V1153I) were detected in three boys with ASD born to unrelated parents with a normal phenotype, and were located within Reelin repeat 1 and 2, respectively [51]. Previous studies suggested that the rs362691 (p.L997V) variant of the *RELN* gene is associated with an increased risk of ASD [51,52]. Persico et al. reported that a polymorphic GGC repeat located in the 5’ untranslated region of the *RELN* gene was associated with ASD [53]. An ethnicity-based subgroup analysis of a meta-analysis found that the single-nucleotide polymorphism (SNP) rs736707 in the *RELN* gene correlated with psychiatric disorders, including ASD, in an Asian group [54]. In an investigation on the relationship between the *RELN* gene and symptoms in children and adolescents with ASD, SNP rs2229864 was linked to a genetic predisposition to ASD, while negative relationships were detected between rs2229864 and symptom-based and developmental characteristics [55]. Previous studies focused on the polymorphisms rs736707, rs362691, and rs2229864 and GGC repeats, but found no correlations with ASD in a meta-analysis [56,57]. Since these findings may be influenced by ethnic groups and sample sizes, further studies are needed to elucidate the relationship between ASD and the *RELN* gene.

### 3.3. AD

AD is a dementia that develops with progressive cognitive impairment and severe neurodegeneration. It is characterized by the extracellular deposition of the amyloid-beta (Aβ) peptide, generated from the β-amyloid precursor protein (APP), and intracellular abnormally hyperphosphorylated tau protein, forming neurofibrillary tangles [58]. The majority of AD patients develop neuropsychiatric symptoms [59]. Previous studies suggested a relationship between the pathophysiology of AD and Reelin signaling [60,61,62]. Reelin may rescue the suppression of LTP and NMDARs induced by the Aβ oligomer [60]. Furthermore, Reelin signaling may prevent the Aβ-induced endocytosis of NMDARs, and SFK activation induced by Reelin has been shown to restore the activity of NMDARs [60]. Conversely, the reduced expression of Reelin enhanced APP processing and amyloid plaque deposition as well as neurofibrillary tangle formation in the hippocampus of aged transgenic AD mice that express the human APP695 gene containing the Swedish (K670N and M671L) and Arctic mutations (E693G) [61]. Reelin expression was found to be up-regulated in the brains of AD patients, while the phosphorylation of Dab1 was decreased, indicating that Reelin signaling is diminished in AD patients. Although a treatment with Aβ increased the expression of Reelin, secreted Reelin was trapped within Aβ aggregates [62]. Accordingly, Aβ may affect the pathological progression of AD by inhibiting the biological activity of Reelin and, ultimately, impairing Reelin signaling [62].

### 3.4. Lissencephaly

Lissencephaly is a severe developmental disorder that is characterized by the lack of development of brain folds and grooves. Hong et al. reported that autosomal recessive lissencephaly with severe abnormalities in the cerebellum, hippocampus, and brainstem was associated with two independent *RELN* mutations identified from British and Saudi Arabian pedigrees. Mutations interfered with the splicing of *RELN* cDNA, leading to low or undetectable amounts of the Reelin protein [63]. Chang et al. identified a homozygous chromosomal inversion, which interrupted the *RELN* gene, in a girl from Turkey who was evaluated for growth and motor retardation. She also had developmental delay, severe hypotonia, seizures, diffuse pachygyria, and severe cerebellar hypoplasia, with a negligible amount of the Reelin protein in her serum [64]. In addition, biallelic splice variants of Dab1 were identified in a patient with mild lissencephaly and cerebellar hypoplasia, and these variants were suggested to affect the highly conserved functional phosphotyrosine-binding domain of Dab1 [65]. Collectively, these findings indicate that marked decreases in Reelin protein expression and Reelin signaling cause lissencephaly.

### 3.5. Mood Disorders

Reelin deficiency has been implicated in the pathophysiology of mood disorders, such as major depression and bipolar disorder. In an immunocytochemical analysis of the hippocampal tissues of postmortem patients with major depression, the number of Reelin-positive cells was consistently lower in subjects with major depression than in controls [66]. Reelin protein expression in patients with major depression was found to be slightly down-regulated in the molecular layer of the dentate gyrus of the hippocampus [67]. A postmortem brain study revealed a significant decrease in Reelin mRNA expression levels in bipolar patients [68]. Furthermore, the number of Reelin-positive cells in the hippocampus was lower in bipolar patients than in controls [66].

As mentioned above, Reelin is associated with several neuropsychiatric disorders. However, future studies are needed to determine the molecular mechanisms of Reelin in synaptic development and function related to these disorders.

## 4. Experimental Animal Models Based on Reelin Dysfunctions

Previous clinical studies reported reductions in Reelin protein and mRNA expression levels in patients with psychiatric disorders. As shown in Table 1, experimental studies on the mechanisms underlying neurological disorders and therapeutic development were conducted using experimental animal models with reduced Reelin expression and functions. In the following section, we review animal models based on Reelin dysfunctions.

### 4.1. Reeler Mice

Jackson *reeler* mice, carrying a 150-kb genomic deletion in the *Reln* gene, are spontaneous mutant mice exhibiting ataxia and are deficient in the Reelin protein [71,72]. In homozygous *reeler* mice, cellular disorganization is observed in the cortical structures of the brain [5]. Neuronal migration is inhibited in the depths of the cortex in the brains of homozygous *reeler* mice in the developmental stage [73]. The levels of MAP2, a dendritic marker, were diminished in the hippocampus of homozygous and heterozygous *reeler* mice. In dissociated hippocampal cultures, the total lengths of the dendrites of homozygous and heterozygous *reeler* mice were significantly shorter than that of wild-type mice. A recombinant Reelin or brain-derived neurotrophic factor (BDNF) treatment ameliorated impaired dendritic growth in the hippocampal neurons of *reeler* mice [10]. Dendritic spine density was found to be reduced in the PFC of heterozygous *reeler* mice. In addition, NMDA-dependent LTP was not induced in the PFC synapses of heterozygous *reeler* mice [30]. Methamphetamine-induced hyperlocomotion was significantly attenuated in *reeler* mice. In addition, locomotor responses to the dopamine D_1_ receptor agonist SKF82958 and dopamine D_2_ receptor agonist quinpirole were decreased in *reeler* mice, suggesting that Reelin plays important roles in dopaminergic functions in the brain [89]. Moreover, GABAergic neurons and their synaptic transmission are altered in neuro-psychiatric disorders; in fact, the expression level of glutamic acid decarboxylase 67, a marker of GABAergic neurons, in the frontal cortex was lower in *reeler* mice than in wild-type mice [90]. The density of parvalbumin neurons was shown to be selectively decreased in the dorsomedial and ventromedial subregions of the intermediate striatum and in the caudal striatum of heterozygous *reeler* mice [91]. The maturation of GABAergic synaptic transmission was altered at layer V/VI in the prelimbic area of the PFC, and the balance between synaptic excitation and inhibition was impaired in *reeler* mice [92]. Some behavioral abnormalities in contextual fear conditioned learning, novel object recognition, and prepulse inhibition tests have been reported between heterozygous *reeler* mice and wild-type mice [30,69,70]. *Reeler* mice have altered susceptibility to drug administration. Social interactions were diminished in heterozygous *reeler* mice by the chronic administration of Δ9-tetrahydrocannabinol (THC), the psychoactive component of cannabis. In addition, a treatment with THC increased the anxiety-like response in female heterozygous *reeler* mice and increased reactivity to aversive situations in male heterozygous *reeler* mice, suggesting that a Reelin deficiency affects behavioral abnormalities caused by psychoactive drugs [93].

In Orleans *reeler* mice, a 220-nucleotide deletion is present in the 3’ region of *Reln* mRNA, and a truncated Reelin protein that terminates within the eighth Reelin repeat is produced. The truncated protein in Orleans *reeler* mice is not secreted extracellularly [72]. Homozygous Orleans *reeler* mice show hyperlocomotion and impaired motor coordination and spatial learning [75]. On the other hand, heterozygous Orleans *reeler* mice exhibit behavioral abnormalities in social behavior and motor learning [74]. Methamphetamine-induced hyperlocomotion and dopamine release in the nucleus accumbens (NAc) were significantly lower in Orleans *reeler* mice than in wild-type mice, suggesting that the function of dopamine release is impaired in Orleans *reeler* mice. The expression levels of GABAergic markers were decreased in the NAc and cerebellum of Orleans *reeler* mice [74]. The expression level of Dab1 in the cerebellum of Orleans *reeler* mice was significantly higher than that in wild-type mice, indicating that Reelin signaling is decreased in Orleans *reeler* mice [49]. Therefore, Orleans *reeler* mice exhibit signaling disorders and behavioral abnormalities, mimicking some of the pathologies of neuropsychiatric disorders, such as schizophrenia.

### 4.2. Maternal Immune Activation Model

Immune activation by maternal infection during pregnancy may increase the risk of neurodevelopmental disorders in offspring [94]. An experimental model of maternal immune activation is the maternal administration of polyinosinic:polycytidylic acid (polyI:C), a synthetic double-stranded RNA analog that mimics viral RNA. An intraperitoneal injection of polyI:C into pregnant mice induced sensory gating deficits, the suppression of exploratory behavior, novel object recognition impairments, and increased anxiety-like behavior in the offspring at adolescence [76,77,78,79,80]. Reelin-expressing cells were reduced in the hippocampus of the offspring by maternal immune activation, particularly in the dentate gyrus of the hippocampus [76,79]. Furthermore, offspring that had received maternal immune activation during pregnancy exhibited a decrease in the immunoreactive area of synaptoporin, a synaptic vesicle marker for hippocampal mossy fibers [79] and reduced postnatal neurogenesis in the dentate gyrus of the hippocampus [76]. Impairments in novel object memory and anxiety-like behavior in the offspring that received maternal immune activation during pregnancy were ameliorated by a stereotaxic microinjection of recombinant Reelin into the hippocampus [79]. Accordingly, Reelin supplementation may exert therapeutic effects on the cognitive and emotional impairments of neurodevelopmental disorders.

### 4.3. Repeated Corticosterone (CORT)-Treated Animal Model

Rats subcutaneously injected with CORT (40 mg/kg) are an experimental animal model of depression, and exhibit depressive-like behavior, impaired memory, and reduced numbers of Reelin-positive cells in the hippocampus [83,84,85,95]. In studies that focused on neuropathological changes, a treatment with CORT impaired hippocampal neurogenesis and reduced dendritic complexity [81,82,83]. The expression levels of GABA_A_ β2/3 receptors in the dentate gyrus subgranular zone of the hippocampus were decreased by the repeated administration of CORT [81,82]. Previous studies investigated whether existing drugs and Reelin replacement attenuated CORT-induced neurological dysfunction. CORT-treated rats exhibited increased immobility and decreased climbing and swimming behaviors in the forced swim test [84,85]. The co-administration of imipramine, a tricyclic antidepressant, prevented these behavioral phenotypes, indicating that imipramine exerts protective effects against CORT-induced depression-like behavior. Furthermore, imipramine also prevented decreases in Reelin expression and dendritic complexity in the hippocampus of rats treated with CORT [85]. In addition, the peripheral administration of the anti-inflammatory drug etanercept, a TNF-α inhibitor, ameliorated CORT-induced impairments in the forced swim, object-location memory, and object-in-place memory tests. Etanercept was shown to restore reductions in hippocampal neurogenesis, Reelin expression, and GABA_A_ β2/3 receptors in CORT-treated rats [81]. Recombinant Reelin infusions into the rat hippocampus protected against CORT-induced memory dysfunctions, increases in depression-like behavior, and impaired hippocampal neurogenesis. These effects of Reelin were inhibited by an injection of the AMPAR antagonist CNQX. Furthermore, Reelin rescued CORT-induced decreases in PSD95, mTOR, phosphorylated mTOR, GABA_A_ β2/3 receptors, GluA1, and GluN2B in the rat brain [82].

### 4.4. Reln-Del

A recent CNV analysis of Japanese schizophrenia patients identified a novel pathogenic deletion (12.6kb) in *RELN* encoding Reelin in *RELN*-del [45]. A male schizophrenia patient with *RELN*-del exhibited positive and negative symptoms, cognitive impairment, and repetitive behavior. He also displayed atrophy of the left cerebral hemisphere, particularly in the frontal and parietal lobes. The amount of the NR6 fragment of Reelin was lower in his serum than in the sera of other patients [74]. *Reln*-del mice, genetically modified C57BL/6J mice that mimic *RELN*-del in the schizophrenia patient, were developed by genome editing with the CRISPR/Cas9 system. Reelin protein expression levels in the heterozygous *Reln*-del mouse brain were reduced to approximately 50% of those in the wild-type mouse brain and were barely detectable in the homozygous *Reln*-del mouse brain [88]. Moreover, Reelin mRNA levels were significantly lower in the heterozygous *Reln*-del brain than in the wild-type brain, suggesting that Reelin protein expression was down-regulated based on lower mRNA levels in *Reln*-del mice [87]. Homozygous *Reln*-del mice show severe brain malformations (cerebellar atrophy, enlarged cerebral ventricles, cerebral dysplasia, and disruption of the dentate gyrus and granule layer), while heterozygous *Reln*-del mice have no major deficits in their brain structure. Reaggregation and neuronal migration were severely altered in cerebellar granule neuronal cultures prepared from homozygous *Reln*-del mice, which may be closely related to cerebellar hypoplasia [88]. In vitro analyses using primary cultured cortical neurons indicated that intracellular Reelin protein levels were lower in *Reln*-del neurons than in wild-type neurons. Reelin proteins secreted into the conditioned medium of cortical neurons were also markedly reduced in *Reln*-del neurons. In contrast, Dab1 expression levels were significantly higher in *Reln*-del neurons than in wild-type neurons, suggesting that Reelin signaling was diminished in *Reln*-del neurons. A shorter neurite length and fewer neurite branch points and dendritic spines in *Reln*-del neurons than in wild-type neurons have also been reported, indicating that the defective formation of neurons and dendrites during neurodevelopment is one of the reasons for structural abnormalities in the brains of *Reln*-del mice [87]. Since the patient with *RELN*-del was a heterozygote, heterozygous *Reln*-del mice were subjected to a comprehensive behavioral analysis. In the three-chamber social interaction test, heterozygous *Reln*-del mice exhibited abnormalities in social novelty, suggesting that *Reln*-del mice partially mimicked schizophrenia-like behavior. However, no impairments were noted in other behavioral tests, including the general locomotor function, open field, elevated plus maze, pre-pulse inhibition, Y-maze, and fear conditioning tests [88]. Cognitive function and flexibility in *Reln*-del mice were evaluated using the touchscreen-based visual discrimination and reversal learning tasks [86], which are highly sensitive for detecting cognitive dysfunction in mice [96]. In these tasks, *Reln*-del mice showed impaired associative learning and behavioral flexibility [86].

Human isogenic induced pluripotent stem cells (hiPSCs) were generated by targeted genome editing to establish the *RELN*-del hiPSCs, and separately differentiated into dopaminergic, glutamatergic, and GABAergic neurons [97,98]. Reelin protein expression and the tyrosine phosphorylation of Dab1 were decreased in isogenic *RELN*-del dopaminergic neurons, suggesting that Reelin signaling was diminished in *RELN*-del cells. In addition, a single-cell trajectory analysis showed a wandering type of migration in *RELN*-del neurons [97]. Gephyrin (postsynaptic marker) and Synapsin I puncta were significantly decreased in isogenic *RELN*-del GABAergic neurons. These findings are similar to those reported in the postmortem brains of schizophrenia patients [99,100], suggesting that the synapse phenotypes of *RELN*-del neurons are general phenotypes of neuropsychiatric disorders [98]. Neurons induced from hiPSC lines carrying congenital *RELN*-del had a shorter dendrite length and decreased synapse number and also lost the directionality of migration [97,98]. These in vitro models using hiPSCs with *RELN*-del are considered to be useful for pathological analyses of neuropsychiatric disorders, such as schizophrenia.

## 5. Effects of Enhancements in Reelin Functions

Previous studies indicated that Reelin supplementation and enhancements in Reelin signaling improve neurological functions. Therefore, Reelin may be a therapeutic target for neuropsychiatric disorders. To investigate the direct effects of Reelin on behavior, mice overexpressing Reelin in forebrain neurons (Reelin-OE) were generated [101]. Reelin-OE mice showed a reduced floating time in the forced swim test in mice treated with chronic CORT, and reduced hyperlocomotion induced by cocaine administration. In addition, PPI deficits induced by a treatment with the NMDAR antagonist, MK-801, were significantly attenuated in Reelin-OE mice [101]. A microinjection of Reelin into the medial PFC prevented MK-801-induced impairments in recognition memory and increases in the number of c-Fos-positive cells, suggesting that Reelin prevented MK-801-induced abnormal neuronal activation [102].

The effects of Reelin overexpression on the pathology of tauopathy were investigated using AD-related mice expressing human mutant Tau (G272V, P301L and R406W), which are called VLW mice [103]. Increases in Tau phosphorylation levels in the hippocampus of VLW mice were reduced by the overexpression of Reelin. In addition, LTP deficits and cognitive impairment in VLW mice were ameliorated by the overexpression of Reelin. These findings suggest that enhancements in Reelin signaling protect against the symptoms of Tau pathology. Therefore, Reelin may be a therapeutic target in AD [103].

## 6. Novel Druggable Targets for Reelin Supplementation Therapy in Neuropsychiatric Disorders

Reelin degradation enzymes may be potential targets to enhance Reelin signaling. A disintegrin and metalloproteinase with thrombospondin motifs-3 (ADAMTS-3) has been identified as the protease that specifically cleaves Reelin at the N-t site in the cerebral cortex and hippocampus (Figure 1) [104]. ADAMTS-3 is expressed in the excitatory neurons of the embryonic and postnatal cerebral cortex and hippocampus, and down-regulates Reelin in the embryonic and postnatal brain. The NR2 fragment, a degradation product of Reelin at N-t, was found to be significantly decreased in the cerebral cortex of ADAMTS-3 KO mice. Dab1 expression levels were also reduced in the cerebral cortex of ADAMTS-3 KO mice, suggesting that Reelin signaling is activated by an ADAMTS-3 deficiency.

We recently proposed a novel concept to enhance Reelin signaling by the inhibition of ADAMTS-3 as a novel treatment for neuropsychiatric disorders [87]. To investigate the effects of the inhibition of ADAMTS-3 on Reelin signaling, we generated a primary culture of cortical neurons from wild-type and heterozygous *Reln*-del mice and performed knockdown experiments on ADAMTS-3 using short hairpin RNAs. Reelin cleavage in conditioned medium was significantly decreased, whereas Dab1 expression was reduced by the knockdown of ADAMTS-3, indicating that Reelin signaling was enhanced in primary cultured cortical neurons prepared from both wild-type and heterozygous *Reln*-del mice. Therefore, the inhibition of ADAMTS-3 may be a candidate for the clinical treatment of neuropsychiatric disorders, such as schizophrenia, by enhancing Reelin signaling in the brain [87].

Tau phosphorylation, which is involved in the aggravation of AD, was found to be decreased in the cerebral cortex of ADAMTS-3 KO mice. An ADAMTS-3 deficiency in excitatory neurons increased the branching and elongation of dendrites in the somatosensory cortex [104]. Moreover, reductions in ADAMTS-3 inhibited the deposition of Aβ in App knock-in mice, an AD animal model [105,106], by enhancing Reelin activity, which suggests the potential of an inhibitor of ADAMTS-3 to prevent the progression of AD [107].

ADAMTS-2, which has similar domain structures and substrate specificity to ADAMTS-3, was also shown to contribute to the N-t cleavage and inactivation of Reelin in the postnatal cerebral cortex and hippocampus [108]. At the mRNA level, ADAMTS-3 is highly expressed in the embryonic cerebral cortex and hippocampus, while ADAMTS-2 and ADAMTS-3 are expressed at similar levels in the postnatal cerebral cortex and hippocampus. Therefore, the inhibition of ADAMTS-2 may also be a target for neuropsychiatric disorders in the adult brain [108]. Further studies on ADAMTS-2 and ADAMTS-3 are needed to elucidate their mechanisms of action in the treatment of neuropsychiatric and neurodegenerative disorders.

## 7. Conclusions

In this review, we discussed the relationships between the neuronal functions of Reelin and neuropsychiatric disorders. Furthermore, we introduced experimental animal models based on Reelin dysfunctions. We showed that enhanced Reelin signaling may ameliorate neurological dysfunctions. The down-regulated expression of Reelin and *RELN* mutations have been reported in patients with neuropsychiatric disorders. Reelin supplementation improved neurological functions and may be a candidate for novel treatments for neuropsychiatric disorders. Since the amount of Reelin in serum fluctuates in psychiatric disorders, it may be used as a marker of illness and an indicator of therapeutic efficacy. However, clinical therapies based on Reelin functions have not yet been applied in practical settings. Further studies are needed to develop treatments that target Reelin functions.

## Figures and Tables

**Figure 1 ijms-23-01829-f001:**
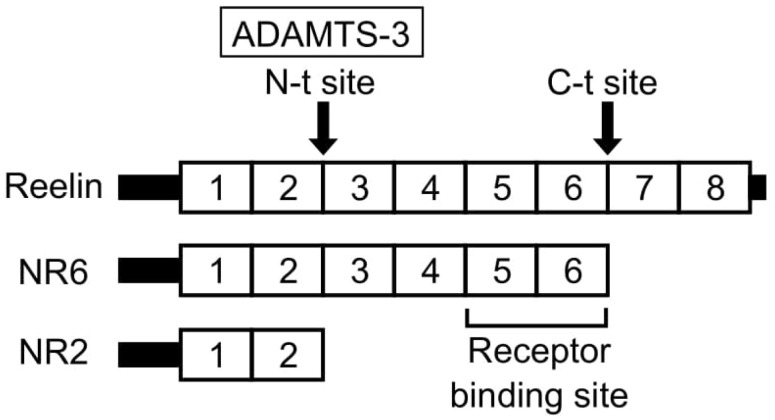
Schematic representation of the Reelin structure and cleavage by a disintegrin and metalloproteinase with thrombospondin motifs-3 (ADAMTS-3).

**Table 1 ijms-23-01829-t001:** Summary of animal models based on Reelin dysfunctions.

Animal Model	Mutation/Treatment	Abnormal Phenotypes	Behavioral Changes	Effects of Reelin Supplementation	References
Jackson *reeler* mice	150-kb genomic deletion in the *Reln* gene	Brain malformation, decreased Reelin protein levels, impaired neurite development, fewer dendritic spines	Impairments in contextual fear conditioned learning, novel object recognition, and prepulse inhibition tests	Elongation of dendrites, enhanced synaptic functions, attenuation of impaired contextual fear conditioned learning and prepulse inhibition	[5,10,30,69,70,71,72,73]
Orleans *reeler* mice	220-nucleotide deletion in *Reln* mRNA	Expressing a truncated Reelin protein that is not secreted extracellularly	(Homozygous) Hyperlocomotion, impairments in motor coordination and spatial learning (Heterozygous) Abnormal social behavior and motor learning	Not available	[72,74,75]
Maternal immune activation model	The offspring of pregnant mice administered polyI:C	Decreased number of Reelin-expressing cells, impaired hippocampal neurogenesis	Sensory gating deficits, suppression of exploratory behavior, impaired novel object recognition, increased anxiety-like behavior	Rescue of impaired novel object memory and anxiety-like behavior	[76,77,78,79,80]
CORT-treated animal model	Rats subcutaneously injected with CORT	Reduction in Reelin-positive cells, impaired hippocampal neurogenesis, decreases in PSD95, mTOR, phosphorylated mTOR, GABA_A_ β2/3 receptors, GluA1, and GluN2B	Increased depressive-like behavior and impaired memory	Attenuation of increased depressive-like behavior and impaired memory	[81,82,83,84,85]
*Reln*-del mice	Mice mimicking *RELN*-del in a schizophrenia patient	Brain malformation, decreased Reelin protein levels, impaired neurite development, fewer dendritic spines	Abnormal social novelty, impaired associative learning and behavioral flexibility	Enhancement in Reelin-Dab1 signaling	[86,87,88]

polyI:C, polyinosinic:polycytidylic acid; CORT, repeated corticosterone; GABA, γ-aminobutyric acid; mTOR, mammalian target of rapamycin; PSD95, postsynaptic density protein 95.

## Data Availability

Not applicable.

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
