# Peer review of "New Strategies for the Treatment of Neuropsychiatric Disorders Based on Reelin Dysfunction"

_ijms, 2022, doi:10.3390/ijms23031829_

Round 1
Reviewer 1 Report
The review by Tsuneura et al. provides new insight for potential treatment of neuropsychiatric disorders related with Reelin dysfunction. The text in general has a nice and clear flow, focused on the relationship between Reelin and several neuropsychiatric disorders, and animal models with Reelin supplementation.
Below some minor comments/suggestions:
- P2, 60-68
Since most well-known Reelin receptors for neuronal migration are ApoER2 and VLDLR, it needs to be explained mainly. Integrin α5β1 has been reported that it does not even directly bind to Reelin.
- P2, 73-74
reeler mice need to be described clearly as described later section (gene deletion) not ‘’lacking Reelin signal’
- P2, 75
Please delete [high-molecular-weight protein]. It doesn’t need to be explained in detail as a marker.
- Several sentences are not clear.
e.g. P2 82-82 Reelin exerted its functions through downstream intracellular Dab1 and SKF signaling cascades and regulated dendritic outgrowth.
P8 352-353 Reelin expression levels in the heterozygous Reln-del mouse brain were reduced to approximately 50% that in the wild-type mouse brain, and were barely detectable in the homozygous Reln-del mouse brain.
P9 388-389 The frequency by which Synapsin I (presynaptic marker) and Homer I (postsynaptic marker) co-localized was slightly lower in isogenic RELN-del glutamatergic neurons.
- P3 102-103
Please add references following [A treatment with Reelin has been shown to significantly increase dendritic spine density in primary cultured hippocampal neurons]
- P3
In case Reelin is associated with several neuropsychiatric disorders, the molecular mechanism of Reelin in synapse development or synaptic functions, whose dysfunction might be also related with neuropsychiatric disorders, need to be explained in detail.
- P7 271-278
It may be more logical for the molecular relationship between reeler and neuropsychiatric disorders by adding descriptions about the alteration of GABAergic neurons and GABAergic synaptic transmission also in several neuropsychiatric disorders.
- P7 303
[Immune activation by maternal infection during pregnancy induces neurodevelopmental disorders in offspring.]
This sentence needs to be corrected. It may increase the risk of neurodevelopmental disorders in offspring….
- P9 382-383
Following sentence needs to be clear. Cells from schizophrenia patient with RELN-del does not need to go through genome editing. Did they study with cells from the patients and separately with cells genetically engineered?
[Isogenic human-induced pluripotent stem cells (iPSCs) derived from the schizophrenia patient with RELN-del were generated by targeted genome editing]
Author Response
Comments to Reviewer 1:
Q1. Since most well-known Reelin receptors for neuronal migration are ApoER2 and VLDLR, it
needs to be explained mainly. Integrin α5β1 has been reported that it does not even directly bind
to Reelin. (P2, 60-68)
A1. According to the comment, we revised as follows:
Moreover, ApoER2 and VLDLR are well known as major receptors involved in neuronal
migration via Reelin signaling [20, 21]. ApoER2, a Reelin-binding receptor, controls several
processes in neuronal migration during cortical development, such as the early stage of radial
migration and the termination of migration [20]. In neonatal ApoER2 knockout (KO) mice,
cortical neurons overmigrate into the marginal zone [20]. (P2, line63-68)
Q2. Reeler mice need to be described clearly as described later section (gene deletion)
not ‘’lacking Reelin signal’. (P2, 73-74)
A2. According to the comment, we also revised it. (P2, line 75)
Q3. Please delete [high-molecular-weight protein]. It doesn’t need to be explained in detail as a
marker. (P2, 75)
A3. We deleted high-molecular-weight protein in original text.
Q4. Several sentences are not clear.
e.g. P2 82-82 Reelin exerted its functions through downstream intracellular Dab1 and SKF
signaling cascades and regulated dendritic outgrowth.
P8 352-353 Reelin expression levels in the heterozygous Reln-del mouse brain were reduced to
approximately 50% that in the wild-type mouse brain, and were barely detectable in the
homozygous Reln-del mouse brain.
P9 388-389 The frequency by which Synapsin I (presynaptic marker) and Homer I (postsynaptic
marker) co-localized was slightly lower in isogenic RELN-del glutamatergic neurons.
A4. Thank you for your suggestion. According to your comments, we revised them as follows:
Kupferman et al. reported that Reelin exerted its functions through downstream intracellular
Dab1 and src-family tyrosine kinase (SFK) signaling cascades and regulated dendritic
outgrowth [23]. (P2, line 81-83)
Reelin protein expression levels in the heterozygous Reln-del mouse brain were reduced to
approximately 50% that in the wild-type mouse brain, and were barely detectable in the
homozygous Reln-del mouse brain [88]. (P8, line 353-355)
We deleted the sentence in original text.
Q5. Please add references following [A treatment with Reelin has been shown to significantly
increase dendritic spine density in primary cultured hippocampal neurons]. (P3, 102-103)
A5. We add reference [29]. (P3, line 104)
Q6. In case Reelin is associated with several neuropsychiatric disorders, the molecular mechanism
of Reelin in synapse development or synaptic functions, whose dysfunction might be also related
with neuropsychiatric disorders, need to be explained in detail. (P3)
A6. We agree your comment, further research are needed for that. Thus, we add a sentence as
follows:
As mentioned above, Reelin is associated with several neuropsychiatric disorders. However,
future studies are needed to be clear the molecular mechanism of Reelin in synapse development
or synaptic functions in the disorders. (P5, line 234-236)
Q7. It may be more logical for the molecular relationship between reeler and neuropsychiatric
disorders by adding descriptions about the alteration of GABAergic neurons and GABAergic
synaptic transmission also in several neuropsychiatric disorders. (P7, 271-278)
A7. We agree your comment, thus we already described GABAnergic dysfunction in Reeler mice.
According to your comment, we revised as follows:
Moreover, GABAergic neurons and its synaptic transmission are also altered in neuropsychiatric
disorders, in fact, the expression level of glutamic acid decarboxylase 67, a marker of GABAergic
neurons, in the frontal cortex was lower in reeler mice than in wild-type mice [90].
Q8. [Immune activation by maternal infection during pregnancy induces neurodevelopmental
disorders in offspring.]
This sentence needs to be corrected. It may increase the risk of neurodevelopmental disorders in
offspring…. (P7, 303)
A7. According to the comment, we revised it as follows:
It may increase the risk of neurodevelopmental disorders in offspring by immune activation
through maternal infection during pregnancy [94]. (P7, line 304-305)
Q9. Following sentence needs to be clear. Cells from schizophrenia patient with RELN-del does
not need to go through genome editing. Did they study with cells from the patients and
separately with cells genetically engineered? (P9, 382-383)
[Isogenic human-induced pluripotent stem cells (iPSCs) derived from the schizophrenia patient
with RELN-del were generated by targeted genome editing]
A9. According to the comment, we revised as follows:
Isogenic human-induced pluripotent stem cells (iPSCs) were generated by targeted genome
editing to establish the RELN-del iPSCs. (P9, line 389-390)

Reviewer 2 Report
Tsuneura et al. described the roles of Reelin in neural functions, and summarized the association of Reelin with several neuropsychiatric disorders. In addition, the authors reviewed various types of animal models mimicking Reelin dysfunctions. Finally, the authors discussed potential therapeutic approaches for these neuropsychiatric disorders associated with Reelin dysfunctions in this review manuscript.
Overall, the manuscript is well-organized and provides valuable information on the function of Reelin in neuropsychiatric disorders, including schizophrenia, autism, and mood disorders.
I have several minor concerns in this manuscript.
1. At the beginning of 3. Reelin and neuropsychiatric disorders (line 137-138), please provide brief background why the authors particularly reviewed the roles of reelin focusing on the neuropsychiatric disorders of schizophrenia, autism, AD, lissencephaly, and mood disorders.
2. Many abbreviations are used without definition in this manuscript. Please define abbreviations when they appear for the first time in the text, and use the defined forms throughout the manuscript.
For example, SKF (line 82), Stk25 (line 85) and STK25 (line 88), MST3 and GCKIII (line 87), and etc.
3. Please specify the abbreviated words at the end of each Table and Figure.
Author Response
Comments to Reviewer 2:
Q1. At the beginning of 3. Reelin and neuropsychiatric disorders (line 137-138), please provide
brief background why the authors particularly reviewed the roles of reelin focusing on the
neuropsychiatric disorders of schizophrenia, autism, AD, lissencephaly, and mood disorders.
A1. Thank you for your kind comment. We add the sentence as follows:
As mentioned above, Reelin has many effects on brain formation and morphological changes in
neuronal network, thus its dysfunction could cause some diseases. In this paragraph, we would
like to introduce the reports showing mutations in the Reelin gene in humans with
neuropsychiatric disorders.
Q2. Many abbreviations are used without definition in this manuscript. Please define
abbreviations when they appear for the first time in the text, and use the defined forms
throughout the manuscript.
For example, SKF (line 82), Stk25 (line 85) and STK25 (line 88), MST3 and GCKIII (line 87),
and etc.
A2. According to the comments, we revised them as follows:
src-family tyrosine kinase (SFK) (P2, line 83)
serine/threonine kinase 25 (STK25) (P2, line 86-87)
mammalian sterile 20-like kinase-3 (MST3) (P2, line 88-89)
germinal center kinase III (GCKIII) (P2, line 89)
Q3. Please specify the abbreviated words at the end of each Table and Figure.
A3. According to the comment, we specified the abbreviated words in Table 1 and Figure 1
